# Epidemiological and phylogenetic analyses of public SARS-CoV-2 data from Malawi

Mwandida Kamba Afuleni[1,2]*, Roberto Cahuantzi[1], Katrina A. Lythgoe[3,4], Atupele Ngina Mulaga[2], Ian Hall[1], Olatunji Johnson[1], Thomas House[1]

1 Department of Mathematics, University of Manchester, Manchester, United Kingdom, 2 Department of Mathematical Sciences, School of Science and Technology, Malawi University of Business and Applied Sciences, Blantyre, Malawi, 3 Big Data Institute, University of Oxford, Oxford, United Kingdom, 4 Department of Biology, University of Oxford, Oxford, United Kingdom

* mwandida.afuleni@postgrad.manchester.ac.uk

**Data availability statement:** All relevant data are within the paper and its Supporting Information files. The primary dataset on COVID-19 daily cases/deaths is available at https://ourworldindata.org/coronavirus,

## Abstract

The COVID-19 pandemic has had varying impacts across different regions, necessitating localised data-driven responses. SARS-CoV-2 was first identified in a person in Wuhan, China, in December 2019 and spread globally within three months. While there were similarities in the pandemic's impact across regions, key differences motivated systematic quantitative analysis of diverse geographical data to inform responses. Malawi reported its first COVID-19 case on 2 April 2020 but had significantly less data than Global North countries to inform its response. Here, we present a modelling analysis of SARS-CoV-2 epidemiology and phylogenetics in Malawi between 2 April 2020 and 19 October 2022. We carried out this analysis using open-source tools and open data on confirmed cases, deaths, geography, demographics, and viral genomics. R was used for data visualisation, while Generalised Additive Models (GAMs) estimated incidence trends, growth rates, and doubling times. Phylogenetic analysis was conducted using IQ-TREE, TreeTime, and interactive tree of life. This analysis identifies five major COVID-19 waves in Malawi, driven by different lineages: (1) Early variants, (2) Beta, (3) Delta, (4) Omicron BA.1, and (5) Other Omicron. While the Alpha variant was present, it did not cause a major wave, likely due to competition from the more infectious Delta variant, since Alpha circulated in Malawi when Beta was phasing out and Delta emerging. Case Fatality Ratios were higher for Delta, and lower for Omicron, than for earlier lineages. Phylogeny reveals separation of the tree into major lineages as would be expected, and early emergence of Omicron, as is consistent with proximity to the likely origin of this variant. Both variant prevalence and overall rates of confirmed cases and confirmed deaths were highly geographically heterogeneous. We suggest that real-time analyses should be considered in Malawi and other countries, where similar computational and data resources are available.

geographic data (shapefiles) is accessed from https://data.humdata.org/dataset/cod-ab-mwi?, COVID-19 case data in districts is available at https://covid19.health.gov.mw/ and the link to the population data is https://malawi.unfpa.org/en/publications/malawi-2018-population-and-housing-census-main-report. The authors cannot legally distribute the sequence data supporting the phylogenetic results of this study however, the primary data files are publicly available at the GISAID website, https://www.gisaid.org/. To download the genomic data, first time users need to create an account then login. After login, go to EpiCoV tab and select search. On location, select Africa/Malawi, then check the virus name box before clicking the download button. The authors confirm that they did not have any special access privileges to the sequence data that others would not have. For any additional data requests, please contact mwandida.afuleni@postgrad.manchester.ac.uk.

**Funding:** MKA was supported by the Schlumberger Foundation–Faculty for the Future. KL was supported by the Royal Society and the Wellcome Trust (107652/Z/15/Z) and the Li Ka Shing Foundation. OJ was supported by the Wellcome Trust (228264/Z/23/Z). KL, IH and TH were supported by the Wellcome Trust (227438/Z/23/Z). RC, IH and TH were supported by the UKRI Impact Acceleration Account (IAA 386). We gratefully acknowledge all data contributors, i.e., the Authors and their Originating laboratories responsible for obtaining the specimens, and their Submitting laboratories for generating the genetic sequence and metadata and sharing via the GISAID Initiative, on which this research is based. The funder had no role in study design and analysis, decision to publish, or preparation of the manuscript.

**Competing interests:** The authors have declared that no competing interests exist.

## Author summary

Malawi detected its first infection with SARS-CoV-2 at the start of April 2020, and like many other countries in the Global South did not have comparable volumes of data to Global North countries to inform its response to the COVID-19 pandemic. Here, we present quantitative analyses of the epidemiology and phylogenetics of SARS-CoV-2 in Malawi using open software and data that can be straightforwardly deployed in other countries and for other pathogens, under similar data availability. We observed five major COVID waves over a period from April 2020 to October 2022, each associated with different variants of SARS-CoV-2, as well as significant geographical heterogeneity. Waves were typically associated with early doubling times of between 7 and 4 days, with the second major wave driven by the Beta variant rather than the Alpha and Gamma variants observed in some other countries. Pylogenetic analysis revealed a temporal tree structure consistent with both major variant structure identified elsewhere, and known epidemiology of major variants.

## Introduction

Africa was the last continent to be hit by COVID-19, and the first case was reported on 14 February 2020 in Egypt. The virus spread across the continent within three months. Lesotho was the last state in Africa to be affected [1]. SARS-CoV-2 spread to all continents, risking many lives and lowering global life expectancy [2].

According to the WHO, SARS-CoV-2 variants have one of six main names: Alpha; Beta; Gamma; Delta; Omicron; and Non Variants of concern (non-VOC), which throughout this study is referred to as 'Other' [3]. Omicron was first identified in South Africa in November 2021, then later spread more rapidly to other countries [3]. Although Omicron is highly transmissible, disease severity is lower than for Delta. Delta emerged before Omicron and was first identified in India in late 2020. In late 2020 in South Africa, the Beta variant was discovered for the first time and it was believed to have many mutations, more severe and caused more hospitalisations and deaths than the other variants [3].

Malawi registered its first COVID-19 case on 2 April 2020 [4], two weeks after pre-emptively declaring a state of national disaster on March 20, 2020 [5] [6]. Following the first confirmed case, the Malawi government put in place COVID-19 restrictions that included; banning international travel, prohibiting public events, closing schools at all levels, decongesting workplaces (where people were encouraged to work from home) and reducing the capacity of public transport [7]. In addition, the government recommended wearing face masks, frequent washing of hands and massive testing of everyone showing signs and symptoms of COVID-19. Furthermore, the state made sure that the public was aware of the pandemic, its symptoms and the preventive measures, by using various communication mediums and vernacular languages [7]. Research shows that by February 2021, SARS-CoV-2 spread across all regions of Malawi, and residents from rural and urban areas were equally exposed to or infected with SARS-CoV-2. However, Ngwira *et al* (2021) in their study carried out from April to October 2020 found that the infection risk was higher among people living in the major cities of Malawi than those living in rural areas. The authors attributed their findings to higher population density in the cities than in the rural areas, implying that the higher the population density, the faster the spread of the virus [8].

We conducted, to our knowledge, the first comprehensive analysis of the disease epidemiology and phylogenetics of COVID-19 in Malawi using public data. We identify lineages

associated with the major waves of COVID-19 and the time of most recent common ancestor (tMRCA) of the variants. Geographical heterogeneity of SARS-CoV-2 in Malawi is revealed, which is one of the important factors to consider when combating an infection outbreak. The paper further describes trends of SARS-CoV-2 confirmed cases and confirmed deaths; contrasts the infectiousness of the variants; and quantifies the infection's case fatality rate for each wave.

## Materials and methods

### Population and study setting

This study used data from Malawi. Malawi is a landlocked country that is bordered by Mozambique to the East, South and Southwest; Tanzania to the north and northeast; and Zambia to the West and Northwest. The country lies between latitudes 9° 22′ S and 17° 03′ S, and longitudes 33° 40′ E and 35° 55′ E. Malawi has a total surface area of 118,484 km$^2$ [9] and a total population size of approximately 18 million [10]. The country is divided into three regions—North, Central, and South—and comprises 28 districts. According to Malawi's 2018 population and housing census the capital city, Lilongwe, which lies in the central region has the highest population proportion of 9.3%. More than 80% of the people live in rural areas and 16% in the urban areas. Half of the Malawi population is 17 years old or below [10]. Malawi is categorised as a Low-Income country [11], with a health system that faces significant challenges, including limited financial and human resources [12]. The health system, which has four levels; community, primary, secondary and tertiary, lacks both financial and human resources [13]. In the year preceding the 2018 household census, the country recorded 6.3 all-cause deaths and 32.8 births per 1,000 persons [10].

As of 26 September 2024, the median age for Malawi, Zambia and Zimbabwe was 18 while the median age for Tanzania and Mozambique was 17 and 16, respectively. The proportion of urban population in Malawi (19%) is much lower than in Zambia (46%), Mozambique (41%), Tanzania (39%) or Zimbabwe (38%) [14]. In the year 2022, the crude birth rate per 1,000 people for Malawi (33) was low compared to that of Zambia (34), Tanzania (36) and Mozambique (36), but higher than that of Zimbabwe (30). All-cause death rate per 1,000 people is 7 for both Malawi and Zambia, which is lower than that of Zimbabwe (9), and Mozambique (9). However, the death rate of 7 per 1,000 people for Malawi is slightly higher than that of Tanzania, which is 6 per 1,000 people [15].

### Data sources

**Confirmed cases and deaths.** SARS-CoV-2 was first identified in Malawi on 2 April 2020. New confirmed cases continue to be reported to date, although the numbers are relatively small. In this study, we use data, S1 Data on daily new confirmed cases of SARS-CoV-2 beginning from 2 April 2020 to 19 October 2022, and daily new deaths, S2 Data beginning from 7 April 2020 to 19 October 2022. The day 19 October 2022 was the end date for this study. This data was extracted from Our World in Data (OWID) COVID-19 database for daily confirmed cases and confirmed deaths available at https://ourworldindata.org/coronavirus [16].

**Genomic data for SARS-CoV-2.** COVID-19 sequence data for Malawi was accessed from an open database, GISAID (the Global Initiative on Sharing All Influenza Data), under EpiCoV. The link to GISAID is https://www.gisaid.org/ [17–19]. The data comprised of 1,436 DNA sequences that were collected from 5 April 2020 to 19 January 2023. We downloaded the file containing the sequences in FAST-All (FASTA) format and other files in TSV and TAR formats containing metadata including collection and submission date, host, additional host,

location, genome, lineage, clade, gender, patient age, sampling strategy and AA substitutions, among others.

**Geographical and demographic data.** For mapping, Shapefiles containing the boundaries of sub-national regions in vector format were obtained from the Humanitarian Data Exchange at https://data.humdata.org/dataset/cod-ab-mwi? [20]. The data, S3 Data on the total SARS-CoV-2 confirmed cases in districts were obtained from the the dashboard of the Ministry of Health-Malawi, available at https://covid19.health.gov.mw/ [21] while the demographic data on population sizes for these regions were obtained from the reports of the Malawi National Statistical Office available at https://malawi.unfpa.org/en/publications/malawi-2018-population-and-housing-census-main-report [10].

## Data analysis

**Case fatality rate (CFR) and growth rate analysis.** We quantify and compare the disease severity amongst the variants, by computing CFR with 95% CrI derived using Bayesian conjugate inference for each variant. The ratio of the CFR's and the uncertainty in this ratio is computed using Monte Carlo approximation methods, and from this percentage differences in CFR between pairs of variants are calculated. We fitted case and death data using R statistical software [22], and the Generalised Additive Models (GAMs) R package [23] with log link function. We used these to calculate growth rate and doubling time following the approach of [24,25], noting that if $s(t)$ is a smoothed estimate of the logarithm of a mean signal, then its time derivative $r(t) = \dot{s}(t)$ is an estimate of the instantaneous growth rate and $\tau_D = \log(2)/r(t)$ is an estimate of the instantaneous doubling time [26]. A doubling time analysis expresses the exponential growth rate of an epidemic by measuring how quickly the cases double over time. A shorter doubling time potentially indicates faster spread and a longer doubling time potentially indicates slower growth and/or containment of the epidemic.

GAMs were applied particularly for their flexibility, reliability and validity when fit to non-linear and real-world data. The confirmed Cases or confirmed deaths were a function of time (day and day-of-week) in the model. Day-of-week was considered because during weekends hospitals only accept and treat emergencies, implying that the number of people who report illness to the hospital during weekends is likely to be less than the number reporting illness during weekdays. This affects the figures of the reported cases and deaths during weekends. Further details on how the model was fit to data as well as calculation of the instantaneous growth rate and doubling time of the infection are found in S1 Text.

**National phylogenetic analysis.** We used viral genomic data to identify SARS-CoV-2 variants, and TreeTime [27] to perform a time-scaled phylogenetic analysis in order to describe the evolution of the virus including the time of the most recent common ancestor (tMRCA) for the variants. Multiple sequence alignment was performed on 1,436 sequences using a program known as multiple alignment using fast Fourier transform (MAFFT) [28], which was implemented in Python. The aligned sequences were later reconstructed into a tree using IQ-TREE [29] software package, which is a stochastic algorithm that builds trees by maximum likelihood methods. The tree was later fit to a calendar and visualised using TreeTime version 0.82 [27]. Sequences in a FASTA file were also categorised according to names that WHO and researchers agreed to give to specific COVID-19 lineages [30]. Five categories were formed as shown in a table in S1 Table. Alignment and tree reconstruction were performed before the trees were visualised in interactive tree of life (iTOL) [31].

**Geographical analysis.** We mapped confirmed cases, total population, and cases per capita using the R package cartography [32]. Sequence alignment and tree reconstruction were repeated for individual variants and were then annotated in iTOL [31] to show the

regions the sequences were collected from and compare the distribution of the variants across the three regions of Malawi. We performed a $\chi^2$ test for the difference in proportions in R using the MASS package [33].

## Results

The study used 88,084 confirmed cases and 2,683 confirmed deaths for the epidemiological analysis, and the geographic data had a total of 86,035 confirmed cases. The genomic data comprised 1,436 sequences.

### Case Fatality Rate (CFR)

The overall CFR for the entire study period is 3.05% (95% CrI: 2.94% – 3.16%). Delta was more lethal, CFR: 4.16% (95% CrI: 3.93% – 4.40%) than the other variants while Omicron had the lowest CFR, 1.46% (95% CrI : 1.32% – 1.61%). Table 1 compares the CFR and their 95% credible intervals for the variants. The posterior distribution of CFRs further confirms that the Delta variant had the highest case fatality rate, while the Omicron variant had the lowest. The full posterior distribution of CFRs for the variants, which indicates overlapping CFRs for the Other and Beta variants, is provided in S1 Fig. Further analysis of the ratios of CFRs, transformed to percentage changes, is shown in Fig 1.

### Trends of COVID-19 confirmed cases and confirmed deaths in Malawi

**COVID-19 reported cases modelling.** As of 19 October 2022, Malawi had a total of 88,064 confirmed cases of SARS-CoV-2 [16]. Fig 2A shows the number of daily cases from 2 April 2020 to 19 October 2022. Five distinct peaks are visible in the plot, representing five

**Table 1. Case Fatality Rate for SARS-Cov-2 variants observed in Malawi**

| Variant | Confirmed deaths | Confirmed cases | CFR (%) | 95% CrI=credible interval |
|---|---|---|---|---|
| Other | 185 | 6,043 | 3.07 | (2.66, 3.53) |
| Beta | 957 | 27,932 | 3.43 | (3.22, 3.65) |
| Delta | 1,160 | 27,871 | 4.16 | (3.93, 4.40) |
| Omicron | 381 | 26,218 | 1.46 | (1.32, 1.61) |

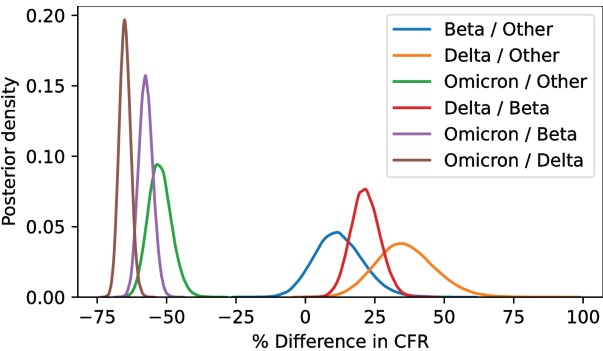

**Fig 1. Case Fatality Ratio Differences.** Posterior densities of the ratios of different strain CFRs, showing high credibility of increase for Delta and decrease for Omicron compared to everything else, but only weak evidence for a difference between Beta and 'other' strains.

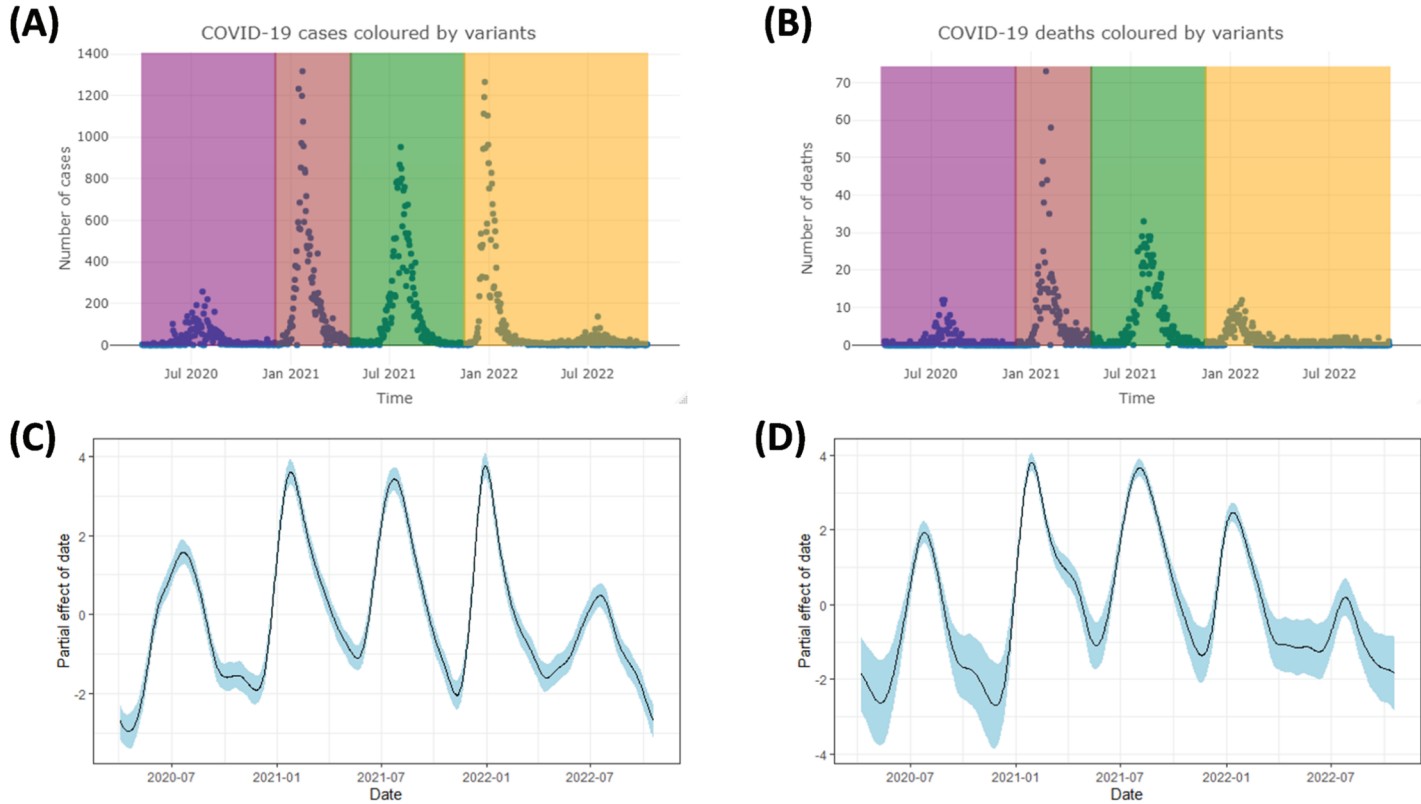

**Fig 2. Trends for SARS-CoV-2 confirmed cases and confirmed deaths.** (A) Daily confirmed cases coloured by variants, Other(purple), Beta(brown), Delta(green) and Omicron(orange). (B) SARS-CoV-2 daily confirmed deaths, coloured by variants, Other (purple), Beta (brown), Delta (green) and Omicron (orange). (C) Partial effect of day on confirmed cases. (D) Partial effect of day on confirmed deaths.

waves of infection, though the final peak is notably brief. On average, 95 people were confirmed infected every day, with 1,316 as the highest observed number of confirmed cases and 0 as the lowest. The variance was $41,200$, which implies that there was much variation in the values for the number of daily confirmed cases; and 95% of the daily number of confirmed cases were below 547. The colours in Fig 2A, each represents a different wave dominated by a COVID-19 variant, which implies that one variant spanned over the last two waves. Dates at which waves started or new colour began were determined by the use of linear discriminant analysis (LDA). In that regard, the Beta wave started on December 3, 2020, the Delta wave on April 21, 2021 and Omicron on November 16, 2021. According to genome data, the first variant identified in Malawi was non-VOC, which predated the Alpha variant and is referred to here as Other. The second was Beta followed by Delta. Omicron was the last wave. Therefore, in Fig 2A, the first peak, to the far left (in purple) is the Other variant, next is Beta (in brown), followed by Delta (in green) and the rest (in orange) is Omicron. The number of confirmed cases peaked during Beta and Omicron waves.

Case data is over-dispersed: the variance to mean ratio is 436, which is considered large. The mean is much greater than the median owing to the presence of few large values of confirmed cases. A Generalized Additive Model (GAM) was fit to the case data using a negative binomial distribution with a log link function. This model accounted for day and day-of-week

effects and is visualised in Fig 2C. The model plot shows the partial effect of day on the confirmed cases and has has five waves; the first three waves are for Other, Beta and Delta, in that order while the last two waves are for Omicron; suggesting that Beta and Omicron were more infectious than Other and Delta variants. The spline in Fig 2C also shows a strong positive relationship between time (day) and confirmed cases in the initial phase of the pandemic. There was a decrease in disease transmission during September and October 2020, followed by a mild increase in November 2020 then a decrease again till early December 2020. This marked the end of the first wave, which was dominated by Other variant. The same December 2020, the Beta variant erupted and reached a peak around January 2021 then decreased drastically over the next four months. It did not take long before the Delta variant appeared, which ran through April to November 2021 and had a peak around July. During the Delta wave, confirmed cases increased with time between June and July, then decreased afterward. Omicron stayed longer than other variants (11 months), confirmed cases fluctuated greatly with time and peaked in January and July 2022.

The model summary results found in S2 Table indicate that the smooth term for the time (day) is statistically significant ($p < 2 \times 10^{-16}$), signifying a clear effect of time whereas the day-of-week predictor is not significant ($p = 0.882$). The model summary confirms that the model has an adequate number of basis functions ($k = 45$), supported by a notable difference between the degree of the curve (4) and the predictor's effective degree of freedom (35), indicating a good fit. The adjusted R-squared value is 82.5%, which considered high. Model diagnostic plots are shown in S2 Fig.

**COVID-19 confirmed deaths modelling.** As of 19 October 2022, $2,683$ COVID-19 confirmed deaths were reported in Malawi. The first death attributed to SARS-CoV-2 in Malawi occurred five days after the first case was reported. Fig 2B provides the number of daily confirmed deaths recorded from 7 April 2020 to 19 October 2022. Maximum and minimum number of confirmed deaths observed were 73 and 0, respectively. On average, 3 people died every day due to COVID–19 and the variance was 44.0922. Ninety-five percent of the values for daily new confirmed deaths were below 15. Colours in Fig 2B represent different variants. Variant of Other (purple), Beta (brown), Delta (green) and Omicron (orange). Beta caused more confirmed deaths than the rest of the variants did. Although Omicron stayed the longest, it caused few confirmed deaths, below 14 per day. More than half of the values, (53.5%) are zeros.

Fig 2D illustrates a GAM model fit to death data using a negative binomial distribution with log link function. There were more COVID-19 confirmed deaths during the second (Beta) and the third (Delta) waves. Delta was less infectious but caused more confirmed deaths, unlike Omicron, which was aggressive but caused few confirmed deaths. Confirmed deaths increased with time in the initial phase of the pandemic, just like the confirmed cases did. The decreasing trend took about a month and a half before a strong positive relationship between time and the number of confirmed deaths was observed from May to July 2020. Later the relationship became negative till November 2020 aside that it became constant for a moment around October 2020. Confirmed deaths during the Beta variant increased rapidly with time from December 2020 and reached a peak around January 2021, then started to drop till May 2021. Confirmed deaths during Delta rose to a high point from June 2021 and peaked August 2021, then the trend declined over the next three months before the Omicron deaths were observed starting from December 2021 to October 2022. Omicron deaths had two peaks, in January and August 2022.

The smooth term, day is significant ($p < 3 \times 10^{-8}$) and the predictor's effective degrees of freedom are 30. Adjusted R-squared = 76%, which is accepted. The intercept is significant unlike the day-of-week. Diagnostic plots are shown in S3 Fig.

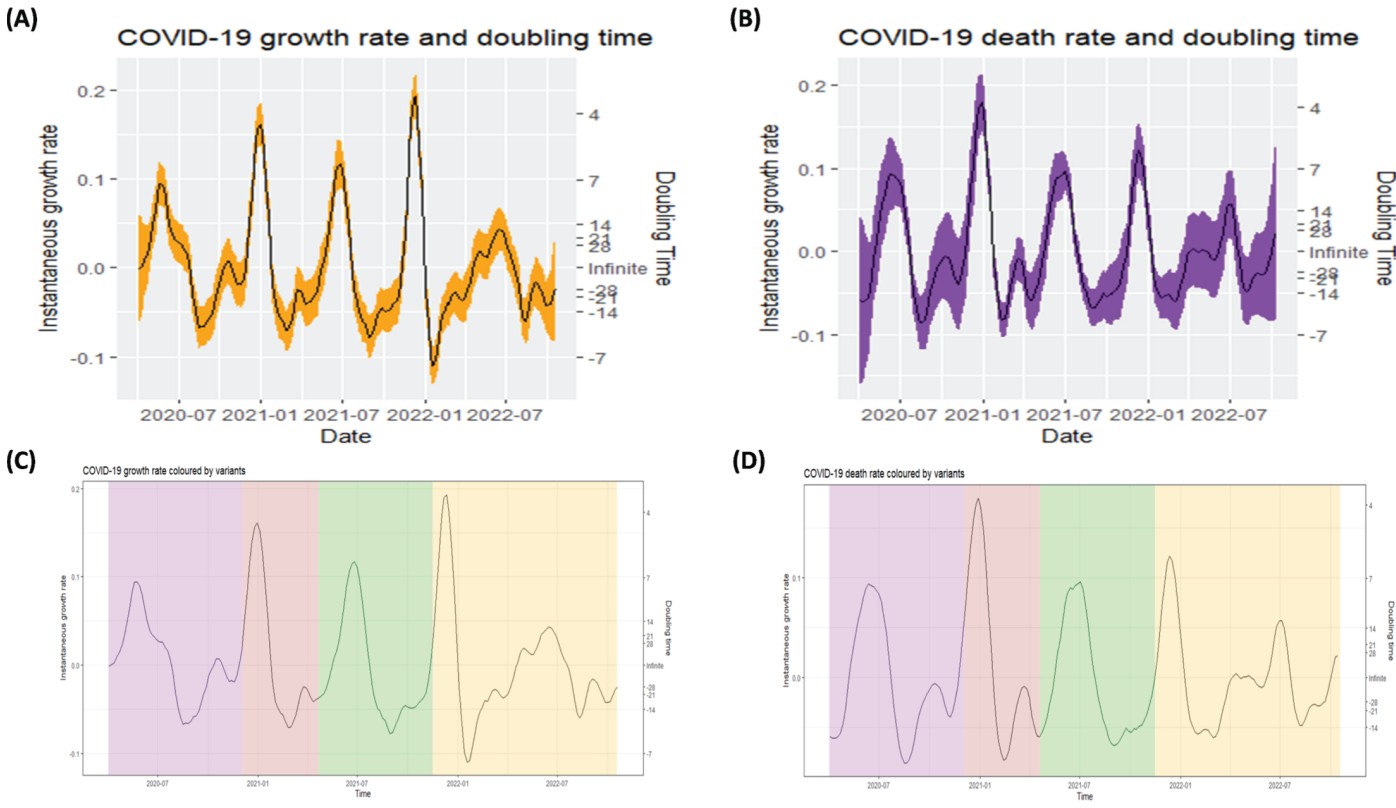

**Fig 3. Growth rate and doubling time.** (A) Growth rate and doubling time curve for confirmed cases. (B) Growth rate and doubling time curve for confirmed deaths. (C) Growth rate for confirmed cases: Other (purple), Beta (brown), Delta (green) and Omicron (orange). (D) Growth rate for confirmed deaths: Other (purple), Beta (brown), Delta (green) and Omicron (orange).

## Growth rate and doubling time

**Growth rate and doubling time of SARS-CoV-2 confirmed cases.** Fig 3A shows a growth rate curve for confirmed cases with 95% confidence interval. When the growth rate curve for confirmed cases is coloured by COVID-19 variants, Fig 3C is produced. The coloured figure allows for comparison of the infection's growth during different waves, which suggests a rapid increase of the infection during Beta and Omicron as compared to Other and Delta waves.

**Growth rate and doubling time of SARS-CoV-2 confirmed deaths.** As shown in Fig 3B, the growth rate for confirmed deaths from 7 April 2020 to 19 October 2022 varied between −0.09 and 0.185 per day, not very different from growth rate for confirmed cases. Unlike in confirmed cases, lowest and highest growth rates for confirmed deaths were observed during Other and Beta variants, respectively. Negative growth rate and doubling time indicate that cumulative confirmed deaths were halving; and doubling otherwise. The general growth rate trend for confirmed deaths appears to be similar to that of confirmed cases. Fig 3D shows death rate curve coloured by variants.

## COVID-19 confirmed cases by district

SARS-CoV-2 confirmed cases across the regions in Malawi are visualised in Fig 4, where Fig 4A shows the total confirmed cases, Fig 4B illustrates the total population and Fig 4C the

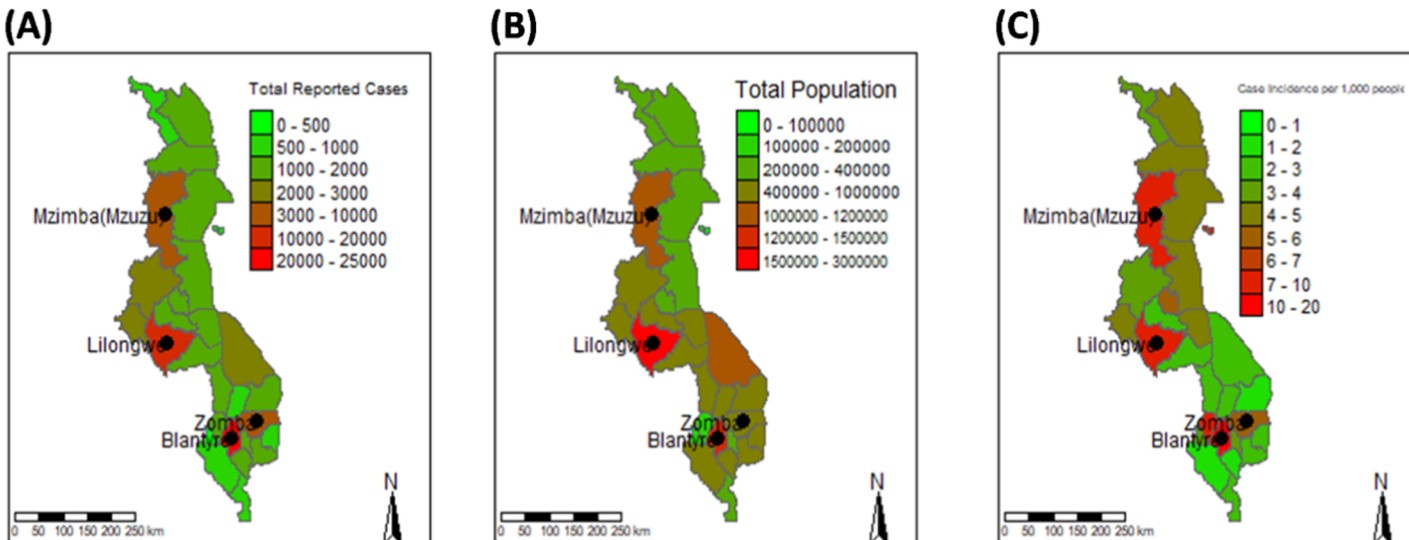

**Fig 4. Maps of confirmed cases, total population and confirmed cases per capita.** (A) Total confirmed cases in districts. (B) Total population in districts. (C) Case incidence per 1, 000 people: Likoma Island and the major cities; Mzuzu (Mzimba), Lilongwe, Blantyre and Zomba have high incidence.

The maps were created by the authors in r programming software version 4.4.1 available at https://www.r-project.org/. The shapefiles are openly available at https://data.humdata.org/dataset/cod-ab-mwi? and the link to the data licence is https://data.humdata.org/faqs/licenses.

case incidence per 1,000 people. Blantyre city in the southern region had the highest number of confirmed cases, 24, 189 followed by the capital city, Lilongwe which had 19, 822 confirmed cases. The least number of total confirmed cases, 118 was observed in the northern region, particularly Likoma island. Three-quarters of the districts realised total confirmed cases below 2, 016. About 64.6% of the total confirmed cases were observed in the four major cities of Malawi including Blantyre, Zomba, Lilongwe and Mzuzu.

## COVID-19 phylogenetics

**Spread of COVID-19 variants across the regions.** More than half of the sequences, 57.3% used in this study were collected from the southern region while 10.2% from the central and 5.8% from the northern part of Malawi however, there was missing information on the region for 26.6% of the sequences. When grouped according to COVID-19 variants as described by WHO, the majority of the sequences belonged to Delta (39.63%), followed by Beta (34.93%), and then Omicron (19.67%). Other and Alpha variants had the least proportions of sequences, 5.35% and 0.42%, respectively. A time tree that resulted from the phylogenetic analysis is shown in Fig 5, where clusters of sequences representing different variants are noticeable. All variants evolved in 2020 however, Omicron took longer to mutate into sub-variants after first emergence than the rest. Comparing the five variants, Beta is closer to Delta and both are closer to Alpha than they are to Omicron. Alpha did not circulate much as depicted from the phylogeny. Fig 5 also shows that Omicron mutated into sub-variants BA.1 and BA.2 where from BA.2 three other sub-variants, BA.4, BA.5 and BQ evolved.

The distribution of SARS-CoV-2 variants across the regions of Malawi is shown in S3 Table. Beta, Delta, Omicron and Other variants spread across all three regions of Malawi while it is hard to conclude for Alpha because of missing information. However, Alpha, Delta and Omicron were more dominant in the southern region.

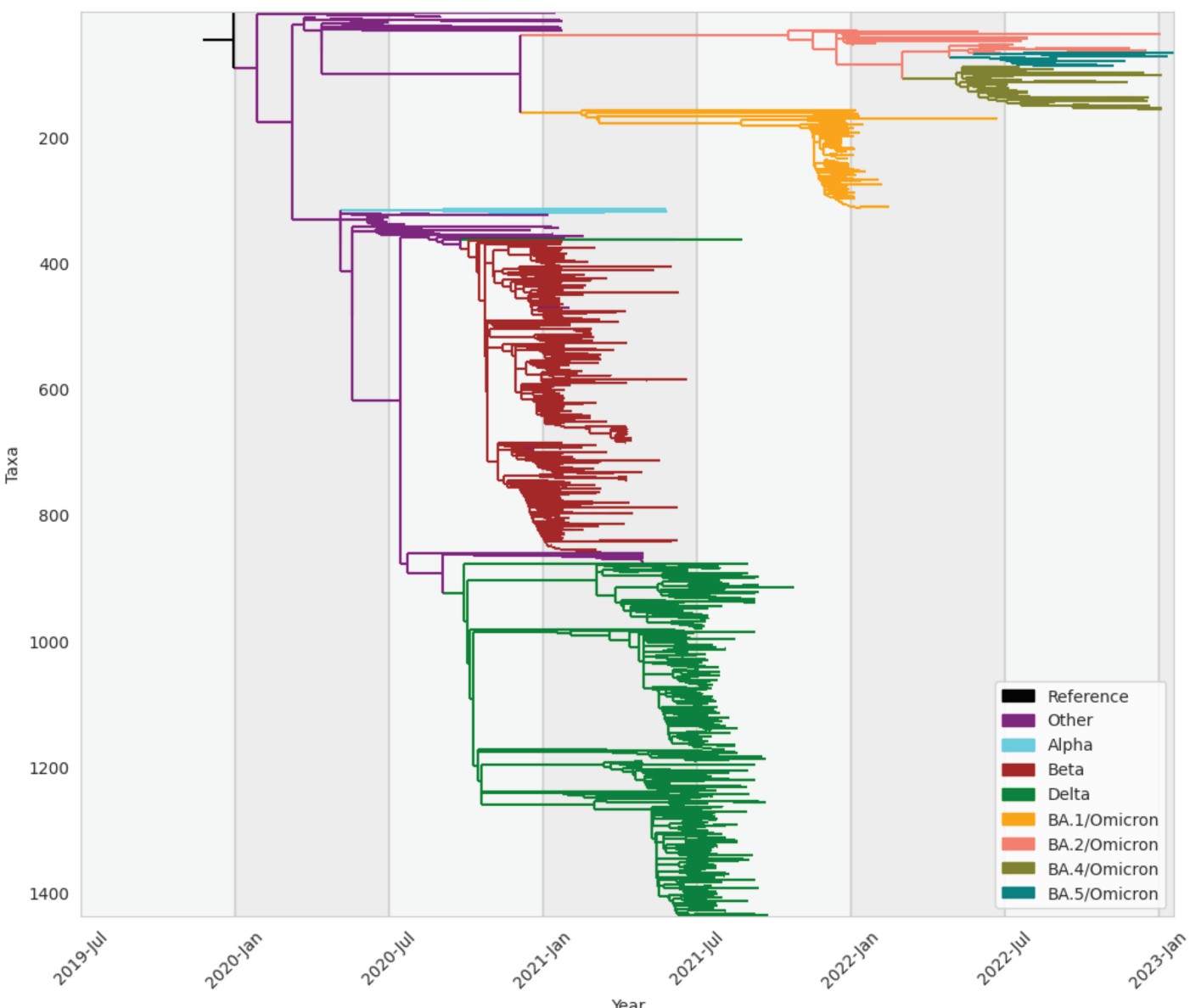

**Fig 5. Maximum likelihood phylogenetic timetree**. Branches coloured according to SARS-CoV-2 Variants: Alpha (blue), Other (purple), Omicron (orange, salmon, olive, teal and magenta), Delta (green) and Beta (brown).

Phylogenetic analysis of each COVID-19 variant revealed that Alpha in Fig 6A, Delta in Fig 6C and Omicron in Fig 6D were more common in the southern region, which is attributed to high population density in the region. Population density is high in the southern region possibly due to the presence of a major commercial city, Blantyre. However, it was not obvious for Beta in Fig 6B and the variant of Other in Fig 6E to identify the regions they were common because 40% and 60% of their sequences respectively, had missing information on the region they were collected from. Despite the absence of information for the majority of the sequences in some variants, it can be observed from the individual trees that there are

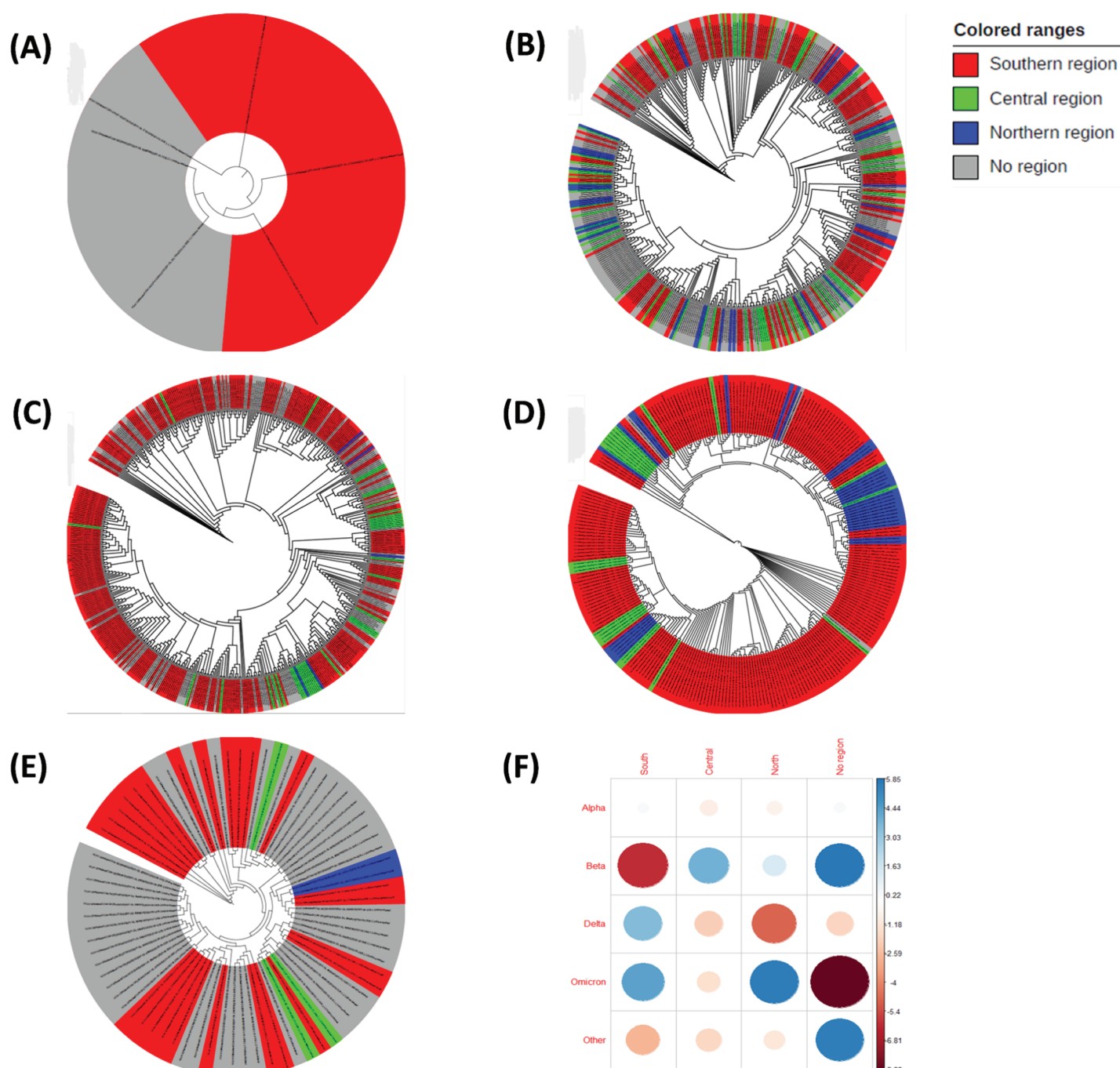

**Fig 6. COVID-19 variants and geographical heterogeneity.** (A) Alpha variant. (B) Beta Variant. (C) Delta variant. (D) Omicron variant. (E) Other variant. (F) Pearson residuals.

possibilities for sequences to be close but found in different regions. Not all related sequences were found in one region i.e. a genome of a sequence found in the North could be very close to another one in the South.

**Chi-square test for the difference in proportions of COVID-19 variants in the regions.** Proportions of COVID-19 variants across the regions and their confidence intervals (CI) are reported in S4 Table. The CI for Alpha could not be computed accurately due to the small number of samples. The contingency table for the spread of COVID-19 variants across the three regions is visualised in the figure found in S4 Fig. The $\chi^2$–test for the difference in proportions found that the proportions of SARS-CoV-2 in the regions were significantly different ($p \approx 0.0005$).

Further analysis of the residuals of the test revealed the cells that contributed to the magnitude of the $\chi^2$–score. As indicated in Fig 6F, with a cut-off point of |3.02|, Beta sequences are significantly less in the southern region than expected however, significantly more than expected in the central region. Delta is significantly more (less) than expected in the southern (northern) region, and Omicron is significantly more than expected in the southern and northern regions. Other variant is significantly less in the southern region than expected. Rated out of 100, cell contribution to the $\chi^2$ score is visualised in S5 Fig. It is clear that Beta in the southern region and that with missing information contribute greatly towards the $\chi^2$ score. Sequences of Other without region information and Omicron in the north and south also contribute highly to the score. This implies that the pandemic hit the northern and southern regions more than the central during the Omicron wave while the southern and northern regions were the least affected during the Beta and Delta waves respectively.

## Discussion

The global and local impacts of COVID-19 have driven the need for extensive research on the pandemic. Many countries were affected across various sectors, such as health, social, and economic. The panic was especially intense in low- and middle-income countries, given their already limited resources [13]. Understanding disease dynamics, viral evolution, and pandemic burden in a low-income country like Malawi is crucial for informing current and future policy interventions. The study finds a case fatality rate of 3% (credible interval: 2.8% to 3.1%) as of October 19, 2022, with an early doubling time ranging from 4 to 7 days. Disease incidence was relatively high in Malawi's four main cities (see Fig 4), likely due to these cities being commercial hubs where most business activities occur. This high incidence may also be attributed to the higher levels of case ascertainment and testing capacity in urban compared to rural areas. The increased incidence on Likoma Island is likely influenced by the low population density. Major variants were identified, and the phylogenetic trees for individual variants showed close sequences appearing in different regions, indicating some inter-regional mobility despite travel restrictions.

Comparing the waves by case numbers, the study found that the first wave, dominated by the 'Other' variant, was the least infectious as seen in Fig 2C and Fig 3C, potentially due to low testing or under-reporting during the early pandemic phase. Vaccination was launched at the time Beta was phasing out and Delta emerging [34] and results show that Delta was less infectious compared to Beta. The reduced infectiousness of the Delta variant, coinciding with the vaccination rollout, suggests a potential impact of vaccination in slowing transmission and establishing herd immunity [35]. However, Omicron was more transmissible despite a substantial portion of the population being vaccinated, though it resulted in fewer deaths compared to Beta and Delta (see Fig 2D and Fig 3D), which agrees with results of the study by Butt (2022) [36] in Qatar and Katella (2022) [3]. Similar results are obtained in this study as shown in the figure demonstrating the posterior distribution of CFR found in S1 Fig, which suggests that Omicron has the lowest CFR while Delta has the highest CFR followed by Beta.

The distribution of COVID-19 variants across the regions of Malawi varied significantly, indicating that the impact of the pandemic was not uniform across the country. Specifically, the Omicron variant had a more severe impact in the southern and northern regions compared to the central region. In contrast, the Beta variant had a relatively limited effect on the southern region, while the Delta variant had a more pronounced impact in the northern region.

Literature shows that the Omicron variant first emerged in South Africa and Botswana in November 2021 [37–39]. Surprisingly, this study finds that Omicron may have first appeared in Malawi as early as December 2020, although it took longer to mutate into multiple sublineages. The study also finds that the highest case incidences are observed in urban areas, primarily in cities. This contrasts with the findings of a spatiotemporal study of COVID-19 in Malawi, which used Geographic Information System (GIS) data and found a higher prevalence in Thyolo District than in the cities of Mzuzu and Zomba [40].

The limitations of this study cannot go unnoticed. One of the weaknesses is that the study used only confirmed cases for mapping case incidence, which could lead to biased results since case ascertainment and testing capacity in the cities is likely to be higher than in the rural areas, considering that over 80% of people live in rural areas. Case incidence in various geographical locations greatly depends on the level of case ascertainment in respective health facilities, which is believed to differ between urban and rural hospitals. Another limitation is that this study used observational data, which limits the insights we can gain. Since the phenomena observed will be driven by complex non-linear disease dynamics, the unpredictable, intricate patterns and interactions that occur within a disease process over time can lead to disproportionately large effects so, in observational statistical work like this it is always hard to give a definitive explanation of all phenomena e.g. in the growth rate and doubling time curves. We hope to return to this in future work. However, the data was collected purposively to manage the epidemic, and therefore, it can still be useful for scientific analysis.

## Conclusion

We analysed the cases, deaths, genomic, and geographical data of COVID-19 in Malawi, identifying five major waves, each dominated by different variants, with Delta and Beta being the most prevalent. While the overall case fatality rate was lower than in high-income countries, it was higher than in other low-income countries. The study also highlighted critical periods of increased transmission and emphasised the need for targeted interventions during these times. The early doubling time of infections was found to be higher than in other regions. The high case incidence observed in cities was likely due to several factors, including population density and testing capacity, among others; therefore, more studies considering these factors are required. Further research is also needed to explore the role of pre-existing immunity, genetic factors, and the impact of early preventive measures on disease outcomes in Malawi. To mitigate future outbreaks, it is crucial to prioritise resource allocation to the northern and southern regions, which were most affected by Omicron. Additionally, ongoing surveillance and rapid response strategies should be emphasised, particularly during periods of high transmission. Given the rapid genetic evolution of SARS-CoV-2, continued vigilance and adaptability in public health strategies are essential for managing future variants and pandemics.

## Code availability

Code for the manuscript is available at https://github.com/MwandieKambaAfuleni/Covid19-Malawi.

## Supporting information

**S1 Text. GAM and growth rate and doubling time calculations.**
(PDF)

**S1 Fig. CFR posterior distribution.**
(PDF)

**S2 Fig. GAM diagnostic for cases.**
(PDF)

**S3 Fig. GAM diagnostic for deaths.**
(PDF)

**S4 Fig. Contingency table for COVID-19 variants.**
(PDF)

**S5 Fig. Cell contribution to Chi-Square score.**
(PDF)

**S1 Table. Grouped lineages.**
(PDF)

**S2 Table. GAM Model summary for cases.**
(PDF)

**S3 Table. SARS-CoV-2 distribution across the regions.**
(PDF)

**S4 Table. Proportions of COVID-19 variants.**
(PDF)

**S1 Data. COVID-19 case data**
(CSV)

**S2 Data. COVID-19 death data.**
(CSV)

**S3 Data. Geographic and population data.**
(CSV)

## Acknowledgments

We gratefully acknowledge all data contributors, i.e., the Authors and their Originating laboratories responsible for obtaining the specimens, and their Submitting laboratories for generating the genetic sequence and metadata and sharing via the GISAID Initiative, on which this research is based.

## Author contributions

**Conceptualization:** Mwandida Kamba Afuleni, Olatunji Johnson, Thomas House.

**Data curation:** Mwandida Kamba Afuleni.

**Formal analysis:** Mwandida Kamba Afuleni, Roberto Cahuantzi, Ian Hall, Olatunji Johnson, Thomas House.

**Investigation:** Mwandida Kamba Afuleni.

**Methodology:** Mwandida Kamba Afuleni, Katrina A. Lythgoe, Atupele Ngina Mulaga, Ian Hall, Olatunji Johnson, Thomas House.

**Project administration:** Thomas House.

**Software:** Thomas House.

**Supervision:** Ian Hall, Olatunji Johnson, Thomas House.

**Visualization:** Mwandida Kamba Afuleni.

**Writing – original draft:** Mwandida Kamba Afuleni.

**Writing – review & editing:** Mwandida Kamba Afuleni, Roberto Cahuantzi, Katrina A. Lythgoe, Atupele Ngina Mulaga, Ian Hall, Olatunji Johnson, Thomas House.

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
