## [Decision Letter · Decision Letter 0]

10 Sep 2024

PGPH-D-24-01510

Epidemiological and phylogenetic analyses of public SARS-CoV-2 data from Malawi

Dear Dr. Afuleni,

Thank you for submitting your manuscript to PLOS Global Public Health. After careful consideration, we feel that it has merit but does not fully meet PLOS Global Public Health’s publication criteria as it currently stands. Therefore, we invite you to submit a revised version of the manuscript that addresses the points raised during the review process.

Thank you for your submission to PLOS Global Public Health. We've received the reviewers' comments on your manuscript. Please address these comments and submit your revised version. 

We look forward to receiving your revised manuscript.

Kind regards,

Charin Modchang, Ph.D.

Academic Editor

Journal Requirements:

1. Please ensure that you provide a single, cohesive .tex source file for your LaTeX revision. You may upload this file as the item type 'LaTeX Source File.' As stated in the PLOS template, your references should be included in your .tex file (not submitted separately as .bib or .bbl). Please also ensure that you are making any formatting changes to both your .tex file and the PDF of your manuscript. If you have any questions, please contact Latex@plos.org. You can find our LaTeX guidelines here:

https://journals.plos.org/globalpublichealth/s/latex

2. Figure 4: please (a) provide a direct link to the base layer of the map (i.e., the country or region border shape) and ensure this is also included in the figure legend; and (b) provide a link to the terms of use / license information for the base layer image or shapefile. We cannot publish proprietary or copyrighted maps (e.g. Google Maps, Mapquest) and the terms of use for your map base layer must be compatible with our CC-BY 4.0 license. 

Additional Editor Comments (if provided):

Thank you for your submission to PLOS Global Public Health. We've received the reviewers' comments on your manuscript. Please address these comments and submit your revised version.

Reviewers' comments:

Reviewer's Responses to Questions

**Comments to the Author**

1. Does this manuscript meet PLOS Global Public Health’s publication criteria? Is the manuscript technically sound, and do the data support the conclusions? The manuscript must describe methodologically and ethically rigorous research with conclusions that are appropriately drawn based on the data presented.

Reviewer #1: Yes

Reviewer #2: Yes

2. Has the statistical analysis been performed appropriately and rigorously?

Reviewer #1: Yes

Reviewer #2: Yes

3. Have the authors made all data underlying the findings in their manuscript fully available (please refer to the Data Availability Statement at the start of the manuscript PDF file)?

Reviewer #1: Yes

Reviewer #2: Yes

4. Is the manuscript presented in an intelligible fashion and written in standard English?

Reviewer #1: Yes

Reviewer #2: Yes

5. Review Comments to the Author

Reviewer #1: Thank you for the opportunity to review this interesting paper where the authors describe the COVID-19 epidemic in Malawi, with a comprehensive overview of epidemiological and phylogenetic analyses. Overall, the methods are very interesting, and the statistical approaches are well described in the main text and supplementary materials. Well done. The writing is generally very good. My main issues are that the authors have not discussed the findings from the three datasets cohesively to paint a more complete portrait of how COVID-19 unfolded in Malawi. Therefore, the added value of using these data in unison is not very clear to me. There is also insufficient discussion of limitations of these data. Finally, despite discussing statistical methods in detail, the reader still doesn’t know why she should be interested in growth rate and doubling time, etc.

I think the authors can work on these issues. I have added some major comments below which I hope the authors can respond to.

Background

- To be succinct and concise, it would help to remove the section on basic information on COVID-19 up to emergence in Wuhan (line 11). Perhaps just start with the situation as it affected Africa as a continent?

- Similarly, consider condensing the (very nice but lengthy) discussion of the different variants to the issue at hand: the emergence in Malawi.

- “Malawi registered its first COVID-19 case on 2 April 2020 [13], two weeks after 39 declaring a state of national disaster on March 20, 2020” (lines 39-40): some readers may be confused as to why Malawi declared the state of emergency before a case was confirmed in its territory. Perhaps add “pre-emptively declaring”

- “A report by the International Food Policy Research Institute (IFPRI) of June 2020, described Malawi’s response to the COVID-19 pandemic as quite satisfactory” (lines 48-9): ‘quite satisfactory’ sounds too informal.

- “This work is to our knowledge the first comprehensive analysis of the disease epidemiology and phylogenetics of COVID-19 in Malawi using public data.” (lines 56-7): State your objectives more clearly, i.e., “We conducted, to our knowledge, the first comprehensive analysis…” As well, what were the sub-objectives for the epidemiological (case/death), phylogenetic, and geographical analyses? How were you trying to link the sets of data to provide a more complete situational analysis?

- Similarly, a criticism of this kind of historical work is that it does not help with the actual management of the epidemic if not done in real-time. What would you say to that? How do you make sense of the overall picture of COVID-19 in Malawi with all these data used in tandem?

- Please remove this, as it isn’t necessary: “The paper has been arranged in a way that the next section describes the data as well as the methods applied to the analyses. After the methods section, results from the analyses are presented and later discussed in the section that follows. The last section concludes the paper by providing the implications of the findings.” (lines 64-7)

Methods

- Under Population and study setting, it would help to compare some of the data with the Southern Africa region, i.e., how does urban/rural proportion and all cause deaths and births compare regionally, or with neighboring countries?

- Under Data sources, please list the exact data sources used for cases and genetic information, i.e., the Malawi Ministry of Health COVID-19 database (just an example). Please remove this statement, which doesn’t add much: “Publicly available datasets on cases, deaths and genome of SARS-CoV-2 were used in this study. This means that the analysis should be completely reproducible by all researchers with Internet access and a modern desktop or laptop personal computer.”

- “In this study, we use data on daily new cases of SARS-CoV-2 beginning from 2 April 2020 to 19 October 2022, and new deaths beginning from 7 April 2020 to 19 October 2022. This was extracted from the Our World in Data (OWID) website [24], and corresponds to a study period of about two and a half years.”: Are these suspect cases or confirmed cases? Please be specific here are everywhere in the manuscript when referring to suspect/confirmed cases and/or deaths. I assume the reason for censoring at 19 October 2022 was this was the end date of this particular analysis, or was there another reason to do so? Also does the OWID actually come from the Ministry of Health? Please acknowledge this as accurately as possible.

- Is FASTA an acronym? This is a general readership journal, so please spell out all acronyms at first use.

- How representative were the sequences available?

- Can you give the reader more information about how these genetic sequences are derived? Who did the genetic analysis, what methods were used etc. I imagine this is fairly generic, but we would like to know some specifics.

- “We fitted case and death data using R statistical package, particularly Generalised Additive Models (GAMs) with log link function”: Should say “R statistical software” and the GAM R package? (is that what you meant?) (lines 115-6)

- I am happy that you looked at growth rate and doubling time, but can you explain to the reader why it is important to look at these quantities in an epidemiological analysis?

- Line 118: What is meant by the smoothed estimate of the logarithm of a mean signal?

- “Day-of-week was considered because reported cases or deaths were seemingly low during weekends.”: Can you explain in plain language why day-of-week effects are important to this analysis?

- Lines 144-6: I think you can remove this to be more concise: “This analysis would be helpful in the future when similar risks arise since provides scientifically proven information to the authorities on whether or not the regions are equally affected, a useful factor to consider when allocating resources to the affected regions.”

Results

- First, give a recap on how many suspect and confirmed cases and sequences were included in the analysis.

- Add “CrI = credible interval” to Table 1

- “The variance was 41, 200 and 95% of the values for daily new cases were below 547.”: Is something wrong with this sentence?

- Figure 3 (panels A, B) needs a more plain language title (not “Derivative of spline arising from GAM”). Add doubling time to the 2nd y-axis in panels D and E. The second bumps in growth rate and doubling time for each variant are interesting. What do they mean?

- Figure 4: since you are emphasizing the increased incidence in cities, can you mark the cities of Blantyre, Zomba, Lilongwe and Mzuzu on the maps? Also check the spelling of incidence.

- Since you are using confirmed cases only for the mapping, how much bias is there towards cities where access to confirmatory analysis were surely more available than the rest of this largely rural country? At the very least, this is a limitation.

- Given that the majority (57.3%) of sequences were collected from the Southern Region, it does not make sense to me to do analysis by region. Please remove these analyses and limit analysis to the whole dataset, or within each region (or justify their use).

Discussion

- I would prefer in the first paragraph of the discussion to list a few of the major findings of the study, rather than restate the methodological steps.

- “Disease incidence was relatively high in the four cities of Malawi (see Fig 4), most likely because they are commercial areas where the majority of businesses take place”: Do you think this may also be because testing was more readily available, therefore increasing the numerator?

- There are many limitations to these kinds of analyses that rely on passively-reported data on infections and deaths, and they certainly impact what you can say with these analysis. One obvious one that comes to mind in the bias in availability of testing (probably more likely in major cities than rural areas?). What about ascertainment of deaths in health facilities and in the communities? There are several others. Please discuss what the limitations are, and how they impact your findings.

- Please don’t add new results into the Discussion/Conclusion (i.e., names of lineages).

- I find the Discussion to be rather dry, as you focus on restating the findings, with some comparisons with other studies that provide evidence to square the findings. I think what is missing is how the three sets of findings and their limitations, provide a more comprehensive picture of the spatial and temporal situation of COVID-19 transmission and deaths in Malawi. Similarly, is there anything that you found is surprising or contrasts with literature from Malawi and the Southern Africa region? I find this kind of discussion essential to this kind of analysis and strongly encourage the authors to expand on this in a few paragraphs.

- The Conclusion, as it stands, is rather wordy. It should be limited to a main overall statement of the major findings.

Reviewer #2: To whom it may concern,

I have carefully reviewed your manuscript on the epidemiology and phylogenetics of SARS-CoV-2 in Malawi. I believe that your work provides valuable insights into the regional impacts of different COVID-19 variants. Your research is commendable for its depth and the integration of various analytical methods, which significantly contribute to understanding the dynamics of the pandemic in a low-income country context.

I recommend that you review the attached file for suggestions on how to further enhance the clarity, coherence, and impact of your manuscript.

6. PLOS authors have the option to publish the peer review history of their article (what does this mean?). If published, this will include your full peer review and any attached files.

**Do you want your identity to be public for this peer review?** For information about this choice, including consent withdrawal, please see our Privacy Policy.

Reviewer #1: No

Reviewer #2: **Yes: **Raquel Muñiz-Salazar

---

## [Decision Letter · Decision Letter 1]

5 Feb 2025

Epidemiological and phylogenetic analyses of public SARS-CoV-2 data from Malawi

PGPH-D-24-01510R1

Dear Ms Afuleni,

We are pleased to inform you that your manuscript 'Epidemiological and phylogenetic analyses of public SARS-CoV-2 data from Malawi' has been provisionally accepted for publication in PLOS Global Public Health.

Best regards,

Charin Modchang, Ph.D.

Academic Editor

Thank you for addressing the reviewers' comments. The reviewers have now recommended the manuscript for publication, confirming that it meets the criteria for publication.

Reviewer Comments (if any, and for reference):

Reviewer's Responses to Questions

**Comments to the Author**

1. If the authors have adequately addressed your comments raised in a previous round of review and you feel that this manuscript is now acceptable for publication, you may indicate that here to bypass the “Comments to the Author” section, enter your conflict of interest statement in the “Confidential to Editor” section, and submit your "Accept" recommendation.

Reviewer #1: All comments have been addressed

Reviewer #2: All comments have been addressed

2. Does this manuscript meet PLOS Global Public Health’s publication criteria? Is the manuscript technically sound, and do the data support the conclusions? The manuscript must describe methodologically and ethically rigorous research with conclusions that are appropriately drawn based on the data presented.

Reviewer #1: Yes

Reviewer #2: Yes

3. Has the statistical analysis been performed appropriately and rigorously?

Reviewer #1: Yes

Reviewer #2: Yes

4. Have the authors made all data underlying the findings in their manuscript fully available (please refer to the Data Availability Statement at the start of the manuscript PDF file)?

Reviewer #1: Yes

Reviewer #2: (No Response)

5. Is the manuscript presented in an intelligible fashion and written in standard English?

Reviewer #1: Yes

Reviewer #2: Yes

6. Review Comments to the Author

Reviewer #1: Second review of PGPH-D-24-01510

Thank you to the authors for attending to the questions I had with very clear answers. I found the responses to be very good, and the narrative reads very well.

The one area that I would improve upon is the description of the purpose of a doubling time analysis (my comment #8):

Under ‘Case fatality rate (CFR) and growth rate analysis’, you wrote: “This provides the authorities with information on the most infectious lineages, for planning of the infection’s preventive and control measures”

I would highly recommend that you be more specific. For instance (ex): “a doubling time analysis expresses the exponential growth rate of an epidemic by measuring how quickly the cases double over time. A shorter doubling time potentially indicates faster spread and a longer doubling time potentially indicates slower growth and/or containment of the epidemic”.

I recommend acceptance of the manuscript, and I leave it to the authors to the make the change I’ve suggested. Well done!

Reviewer #2: All comments have been appropriately addressed.

7. PLOS authors have the option to publish the peer review history of their article (what does this mean?). If published, this will include your full peer review and any attached files.

**Do you want your identity to be public for this peer review?** For information about this choice, including consent withdrawal, please see our Privacy Policy.

Reviewer #1: **Yes: **Ruwan Ratnayake

Reviewer #2: **Yes: **Raquel Muñiz-Salazar
